# Application of Digital Holographic Microscopy to Analyze Changes in T-Cell Morphology in Response to Bacterial Challenge

**DOI:** 10.3390/cells12050762

**Published:** 2023-02-27

**Authors:** Kari Lavinia vom Werth, Björn Kemper, Stefanie Kampmeier, Alexander Mellmann

**Affiliations:** 1Institute of Hygiene, University Hospital Münster, 48149 Münster, Germany; 2Biomedical Technology Center of the Medical Faculty, University of Münster, 48149 Münster, Germany

**Keywords:** digital holographic microscopy, sepsis, bacteria, T-cells, cell morphology

## Abstract

Quantitative phase imaging (QPI) is a non-invasive, label-free technique used to detect aberrant cell morphologies caused by disease, thus providing a useful diagnostic approach. Here, we evaluated the potential of QPI to differentiate specific morphological changes in human primary T-cells exposed to various bacterial species and strains. Cells were challenged with sterile bacterial determinants, i.e., membrane vesicles or culture supernatants, derived from different Gram-positive and Gram-negative bacteria. Timelapse QPI by digital holographic microscopy (DHM) was applied to capture changes in T-cell morphology over time. After numerical reconstruction and image segmentation, we calculated single cell area, circularity and mean phase contrast. Upon bacterial challenge, T-cells underwent rapid morphological changes such as cell shrinkage, alterations of mean phase contrast and loss of cell integrity. Time course and intensity of this response varied between both different species and strains. The strongest effect was observed for treatment with *S. aureus*-derived culture supernatants that led to complete lysis of the cells. Furthermore, cell shrinkage and loss of circular shape was stronger in Gram-negative than in Gram-positive bacteria. Additionally, T-cell response to bacterial virulence factors was concentration-dependent, as decreases in cellular area and circularity were enhanced with increasing concentrations of bacterial determinants. Our findings clearly indicate that T-cell response to bacterial stress depends on the causative pathogen, and specific morphological alterations can be detected using DHM.

## 1. Introduction

Bacterial infections can quickly progress into sepsis, a life-threatening condition caused by a dysregulated host response [1]. Sepsis incidence and mortality remain high with 49 million cases and 11 million deaths per year worldwide [2]. Late diagnosis and delayed or inappropriate treatment are major contributors to these fatality rates. Therefore, different biomarkers, e.g., cytokines, chemokines or acute-phase proteins, have been proposed to facilitate identification of patients at high risk for subsequent organ failure [3]. However, sepsis is a highly complex and heterogeneous syndrome; hence, no single biomarker had sufficient sensitivity and specificity to reliably detect septic patients [4]. In addition to singular molecules as biomarker, analysis of immune cell morphology came into focus as a potential diagnostic approach. Numerous studies using automated hematology analyzers to measure leucocyte volume, conductivity and scatter (VCS) reported changes in monocyte and neutrophil morphology in septic patients compared to non-infected or healthy controls [5,6,7,8]. Variations in cell size, shape and composition correlated with sepsis severity and prognosis and allowed for monitoring of treatment efficacy [9].

Quantitative phase imaging (QPI) is an emerging method in the context of clinical diagnostics [10,11]. Digital holographic microscopy (DHM) [12], which is a variant of QPI, enables the determination of morphological and physical parameters of living cells. It requires neither fixation nor staining or labeling procedures, making it a simple, fast and non-invasive imaging technique that is suitable for long-term measurements. Thereby, DHM provides a promising tool for the diagnosis of diseases that are associated with altered cell morphologies. Potential applications that have been reported thus far include identification of cancer cells [13,14], sickle cell disease [15] and infectious diseases such as malaria [16] or COVID-19 [17].

Since sepsis is also associated with morphological changes in circulating immune cells [18], we aimed to investigate the potential of DHM in sepsis diagnosis. Previous studies showed that DHM is a suitable method to monitor changes in cell morphology in response to bacterial toxins over time [19,20]. In this study, we focused on T-cells as part of the adaptive immune system. T-cell immunity strongly depends on the invading pathogen with great variations not only between different bacterial species but between also strains within the same species [21,22,23]. Based on this context, we hypothesize that T-cells under bacterial stress exhibit changes in cell morphology that are specific for the causative bacteria. Therefore, we chose different Gram-negative (*Escherichia coli*, *Klebsiella pneumoniae*) and Gram-positive (*Staphylococcus aureus*, *Streptococcus pneumoniae*, *Enterococcus faecium* and *E. faecalis*) strains that are frequently detected in septic patients [24,25]. T-cells were either exposed to sterile culture supernatants, bacterial membrane vesicles (MVs) or living bacteria. Both Gram-positive and Gram-negative bacteria produce a number of virulence factors that enable them to interfere with the immune response. The systemic spread of such microbial toxins is a key driver of disease progression and contributes to the outcome of sepsis patients [26,27]. Additionally, MVs are used by many pathogenic bacteria to deliver toxins and other biomolecules into host cells [28]. By entering the systemic circulation, MVs are able to spread throughout the body and interfere with the immune system, thereby contributing to disease progression [29,30,31,32].

Ultimately, we want to find whether the analysis of T-cell morphology using DHM offers a new approach to support early sepsis diagnosis and prediction of the causative agent. Starting an appropriate treatment as early as possible is essential to improve the patient’s outcome and decrease mortality among septic patients.

## 2. Materials and Methods

### 2.1. T-Cell Isolation

Whole blood samples were collected from healthy volunteers after written informed consent was obtained. T-cells were isolated by negative selection using the RosetteSep Human T-cell Enrichment Cocktail (StemCell Technologies, Cologne, Germany) according to the manufacturer’s instructions. Following a density gradient centrifugation with Lymphocyte Separation Medium (Promocell, Heidelberg, Germany) at 800 g for 30 min, the cells were washed twice with PBS (Sigma-Aldrich, Darmstadt, Germany) supplemented with 2% bovine serum albumin (BSA; Serva, Heidelberg, Germany). Finally, the cells were cultured in Roswell Park Memorial Institute 1640 medium (RPMI; Sigma-Aldrich, Darmstadt, Germany) supplemented with 10% fetal calf serum (FCS; PAA, Pasching, Austria) and 2 mM ultraglutamine (Lonza, Cologne, Germany) in a humidified atmosphere at 37 °C and 5% CO_2_.

### 2.2. Bacterial Strains and Culture Conditions

The bacterial strains used in this study are listed in Table 1. *E. coli*, *S. aureus*, and *K. pneumoniae* strains were grown in lysogeny broth (LB; Roth, Karlsruhe, Germany) while *E. faecium*/*faecalis* and *S. pneumoniae* strains were grown in brain heart infusion (BHI) medium (Roth, Karlsruhe, Germany).

### 2.3. Preparation of Bacterial MVs

Bacterial MVs were isolated from overnight cultures as described previously [40]. *E. coli*, *S. aureus*, *E. faecium/faecalis* and *K. pneumoniae* strains were grown at 37 °C with shaking at 180 rpm. *S. pneumoniae* strains were grown stationary at 37 °C and 5% CO_2_. Bacteria were removed from the overnight cultures by centrifugation (5600× *g*, 20 min, 4 °C) and sterile filtration through 0.22 µm pore-size filters. Subsequently, MVs were collected by ultracentrifugation (235,000× *g*, 2 h, 4 °C) in a 45 Ti rotor (Beckman Coulter, Krefeld, Germany), and the pellet was resuspended in 20 mM TRIS-HCl (pH 8.0). The MV suspensions were stored at 4 °C and used for experiments within three months after preparation.

Nanoparticle tracking analysis (NTA) was performed with a NanoSight NS300 instrument (Malvern Panalytical, Kassel, Germany) to determine the MV size and concentration. For every sample, five videos of 60 s were recorded, and particle size distribution and concentration were analyzed by NanoSight NTA 3.4 software (version 3.4.4; Malvern Panalytical, Kassel, Germany). All measurements were performed under constant flow with temperature control at 25 °C. When T-cells were exposed to bacterial MVs, the MV suspension was diluted in cell culture medium (RPMI + 10% FCS + 2 mM ultraglutamine) to a final concentration of 2 × 10^9^ particle/mL.

### 2.4. Preparation of Sterile Culture Supernatants

For preparation of sterile culture supernatants, bacteria were grown in cell culture medium (RPMI + 10% FCS + 2 mM ultraglutamine) overnight at 37 °C and 180 rpm (*E. coli*, *S. aureus*, *E. faecium/faecalis*, *K. pneumoniae*) or 37 °C and 5% CO_2_ (*S. pneumoniae*). Bacteria were sedimented by centrifugation (5600× *g*, 5 min, RT), and the supernatant was passed through 0.22 µm pore-size filters (Corning, Wiesbaden, Germany). Sterile supernatants were freshly prepared prior to every experiment, and T-cells were treated with different concentrations between 5% and 100% (*v*/*v*).

### 2.5. In Vitro Infection

Bacteria were grown in overnight cultures at 37 °C with shaking at 180 rpm. To prepare the infection, the cultures were centrifuged at 5600× *g* and RT for 5 min. The pellet was resuspended in PBS and adjusted to an optical density at 600 nm (OD600) of 1 in PBS (corresponding to 5 × 10^8^ colony forming units (CFU)/mL). T-cells were seeded at a density of 4 × 10^5^ cells/mL and infected with different multiplicities of infection (MOI) ranging from 1 to 10. Four hours post infection (p.i.), 2 µg/mL Lysostaphin (Sigma-Aldrich, Darmstadt, Germany) was added to prevent overgrowth of the bacteria, and timelapse DHM was started. As control, cells were treated in the same way but without addition of bacteria.

### 2.6. Digital Holographic Microscopy (DHM)

Timelapse QPI with DHM was performed with a fiber optic Mach–Zehnder DHM setup via a 20× microscope objective (Zeiss, Jena, Germany) as previously described [23] with minor modifications. Briefly, off-axis holograms were captured automatically every three minutes using a single longitudinal mode laser (Cobolt 06-DPL, λ = 532 nm, 25 mW; Cobolt AB, Solna, Sweden) with an exposure time of 0.15 ms. During acquisition, the sample illumination light was modulated by an electrically focus tunable lens (ETL; Optotune, Dietikon, Switzerland) while series of digital off-axis holograms were recorded to reduce image disturbances caused by the coherence properties of the applied laser light [41]. Quantitative phase contrast images were numerically reconstructed as previously reported [42,43] using custom-built software implemented in python 3.7.

For DHM measurements, T-cells were seeded in 4-well Ph+ µ-slides (ibidi, Gräfelfing, Germany) at a density of 4 × 10^5^ cells/mL, and the slides were sealed with anti-evaporation oil (ibidi, Gräfelfing, Germany). Holograms were captured every three minutes at three different positions for every treatment to record at least 100 cells per time point for a reliable evaluation [44]. Subsequently, quantitative phase images were reconstructed, and further analysis including segmentation and determination of morphological parameters was performed with FIJI software (version 2.3.0/1.53f51) [45]. Morphological parameters such as single cell area, perimeter, circularity and phase contrast values were calculated for every measuring point as mean across all detected cells. Three or more independent biological replicates were performed for every treatment using T-cells obtained from different donors. Results are presented as mean ± SD.

## 3. Results

### 3.1. Strain- and Species-Dependent Morphological Changes in Reaction to Bacterial Culture Supernatants

To investigate the effect of secreted virulence factors on T-cell morphology, human primary T-cells were exposed to sterile culture supernatants of bacterial overnight cultures, and the cellular response was monitored with DHM. Ten hours after the addition of culture supernatants, we observed distinct T-cell shapes, depending on the causative bacteria (Figure 1). In addition to pseudo-colored phase contrast images (Figure 1A), cross-sections through single cells (Figure 1B) are shown to clearly outline the different cell morphologies. *S. aureus* supernatant elicited the strongest effect with entire lysis of the cells. *S. pneumoniae* was the only species leading to an increase in cellular phase contrast while cells treated with *K. pneumoniae* supernatant showed overall decreased phase contrast values. For *E. faecalis* and *E. coli* supernatants, we observed similar morphological changes. Small compartments with higher phase contrast formed inside the cells, whereas in non-treated control cells, the phase contrast values were evenly distributed.

Applying timelapse DHM with image acquisition every three minutes, we were able to quantify changes in cell morphology over time. We included two to three different strains for every species to analyze not only species-specific but also strain-specific differences. The overall pattern of morphological changes varied between the different species but not the strains (Figure 2). Both strains of *S. pneumoniae*, for instance, led to an increase in mean phase contrast and reduction of cell area while the circular shape, quantified by the parameter circularity, was retained. In contrast, supernatant derived from *E. coli* induced a decrease in the cell area and circularity, whereas the mean phase contrast remained constant. The strongest effects were observed for *S. aureus* supernatants that led to rapid lysis of the cells. For that reason, the graphs of strain 6850 and USA300 were cut after 2.5 and 3 h, respectively. Even with dilution down to 5% (*v*/*v*), *S. aureus* supernatants induced a strong decrease in T-cell area (Appendix A). *S. pneumoniae* supernatants caused minor morphological changes at low concentrations, while we did not observe any effect of the remaining species.

Although the response pattern within one species was similar, different strains could be distinguished by the intensity and time course of the morphological changes (Figure 2). The pathogenic *E. coli* strain IHE3034, for example, led to a stronger decrease in circularity than the commensal strain MG1655. Taken together, all strains used in this study induced specific morphological changes and differed from each other in at least one of the analyzed parameters.

### 3.2. T-Cell Response to S. aureus MVs Depends on the Strain

In addition to soluble virulence factors, bacteria release MVs during their normal growth that contain a range of biological cargo including nucleic acids, proteins, enzymes and toxins. MVs can stimulate the innate as well as the adaptive immune response, thereby contributing to pathogenesis within the host [46]. We therefore exposed T-cells to bacterial MVs and monitored changes in single-cell area, circularity and mean phase contrast with timelapse DHM. Only MVs derived from *S. aureus* induced a cellular response (Figure 3). Pseudo-colored phase contrast images clearly show a loss of cell integrity (Figure 3B). This observation is quantified by a decrease in single cell area and circularity score (Figure 3A). In line with the supernatant treatment, the intensity of the cellular response to MVs varied between the different strains. *S. aureus* strain 6850- and USA300-derived MVs induced the most rapid and strongest effects with lysis of the cells after 3 and 7 h, respectively (Figure 3). In contrast, we did not observe any effects on cell morphology when T-cells were treated with MVs derived from *S. pneumoniae*, *E. faecium*, *E. faecalis*, *E. coli* or *K. pneumoniae* (Appendix A).

### 3.3. Cellular Changes in Response to Bacterial Stress Are Concentration-Dependent

It was shown that alterations of neutrophil morphology and motility in sepsis patients correlate with disease severity [9]. Thus, we hypothesized that the T-cell response to living bacteria or bacterial virulence factors might also be concentration-dependent. To test this hypothesis, T-cells were treated with different concentrations of sterile culture supernatant (Figure 4A,C) or were infected with increasing MOIs of living *S. aureus* strain 6850 (Figure 4B). The cellular response to sterile culture supernatant was considerably stronger compared to live bacteria. Even at low supernatant concentrations of 5% (*v*/*v*), T-cell area as well as circularity were rapidly reduced, and increasing the concentration enhanced this effect (Figure 4A). After infection with living bacteria, cell area and circularity also decreased but to a lesser extent (Figure 4B). In line with the previous results, these changes were also concentration-dependent, and a higher bacterial load led to a stronger T-cell response.

## 4. Discussion

DHM is a minimally invasive imaging technique that can give information about the cell number and different morphological parameters including area, thickness, volume or shape [12]. In contrast to other staining-based methods, DHM is time- and cost-saving and overcomes several limitations including phototoxicity or photobleaching. The samples are only exposed to low laser light intensities, allowing for long-term measurements without affecting cell viability. Various diseases are associated with altered cell morphology and motility. Analysis of cellular morphological changes might therefore provide a useful diagnostic tool in the clinical context [10].

In our study, we aimed to investigate the potential of DHM to monitor changes in cell morphology in response to bacterial stress. Furthermore, we were interested in whether differences between various causative agents can be captured. Here, we focused on adaptive immune cells, i.e., T-lymphocytes, as the T-cell response to bacterial pathogens greatly varies depending on the causative agent [21,23]. Primary T-cells were isolated from healthy volunteers and were exposed to bacterial determinants derived from different Gram-negative and Gram-positive species that are frequently detected in septic patients. Additionally, we used multiple strains of the same species to analyze not only species- but also strain-dependent differences. Timelapse DHM was applied to monitor the changes in cell morphology over time.

Pathogenic bacteria produce and secrete a broad range of virulence factors that enable them to evade the immune response, disseminate within the host and cause disease [47]. Bacterial toxins bind to specific receptors on the cell surface and are able to trigger a dysregulated host response, thereby contributing to the progression of an infection to sepsis [26,27,48]. We therefore treated primary T-cells with sterile supernatants from bacterial overnight cultures to analyze the effect of secreted virulence factors on the cell morphology. We observed large variations between the different species and strains (Figure 1 and Figure 2). Representative images 10 h after exposure to culture supernatants show clearly different cell shapes and phase contrast distribution at this time point (Figure 1). Cross-sections through single cells (Figure 1B) are depicted to visualize the differences in cell morphology. Using numerically reconstructed phase contrast images, we quantified changes in single-cell area, circularity and mean phase contrast per cell over time (Figure 2). *S. aureus* strain 6850 and USA300 caused the strongest effects, with entire loss of cell integrity. Cell area and circularity rapidly decreased, and already a few hours after exposure, we were not able to detect any cells. In contrast, all other strains used in this study also induced a reduction of the cell area, but the cells were not completely lysed. The extent of cell shrinkage varied between different bacterial species but not between strains. Strain-specific differences could be observed regarding the cell shape. The extent to which the circularity of single cells declined differed not only between different species but also between strains of the same species. Supernatants from both *S. pneumoniae* strains induced an increase in the mean phase contrast, whereas this parameter was not affected or did not decrease in all other species investigated. Taken together, we observed specific patterns of cellular changes that depended on both the bacterial species and strain. Bacteria are not equally pathogenic since every strain expresses a specific set of virulence factors leading to diverse cellular reactions [21,23,49]. *S. aureus*, as an example, produces various toxins that are able to induce T-cell death, and especially, the α-toxin has a strong cytotoxic effect on T-lymphocytes [50,51]. The expression of these toxins varies greatly between different *S. aureus* strains, thereby influencing the intensity of the cytotoxic effect [49,52]. Furthermore, different bacteria trigger different cell death pathways that are characterized by specific morphological features [53,54]. In a recent study, it was shown that DHM in combination with a deep learning algorithm is able to distinguish apoptotic from necrotic cells [55]. However, it is also known that the immune response not only depends on the invading pathogen but also on different host factors [56]. To address this issue, every treatment was performed at least three times, with T-cells obtained from different donors. Therefore, we can assume that the different observed effects of bacterial virulence factors on T-cell morphology do not result from donor-specific responses. Based on this context, changes in the cell morphology analyzed by DHM might give information about the causative agent.

Gram-negative as well as Gram-positive bacteria produce MVs that contain a broad range of virulence factors and are able to interfere with the immune system [57]. We therefore aimed to investigate if we can confirm our previous results of a strain-specific T-cell response by using MVs isolated from overnight cultures. Previous studies with MVs derived from different bacterial species already showed a dose-dependent cytotoxic effect on various cell types [19,58,59]. The physiological concentration of MVs during bacterial infection has not been reported thus farm although MVs were detected inside the bloodstream [30,60]. For our experiments, we therefore chose a concentration that is in the range of other studies in this research field. Only MVs derived from *S. aureus* strains induced changes in T-cell morphology (Figure 3 and Appendix A). In line with our previous observations, strain 6850 and USA300 caused a strong and fast reduction of cell area and entire loss of cell integrity while strain ST398 had a weaker impact on cell morphology (Figure 3A). *S. aureus* MVs contain biologically active α-toxin that strongly contributes to the cytotoxic effect [58]. In addition, MVs are equipped with numerous other toxins, nucleic acids, proteins and enzymes [61]. Differences regarding the extent of cytotoxicity might therefore result from variations in the MV proteome [62]. In contrast, neither *S. pneumoniae* nor *E. faecalis/faecium*, *E. coli* or *K. pneumoniae* derived MVs were able to induce alterations of T-cell morphology (Appendix A), which is in line with previous studies [23,63,64].

Changes in neutrophil motility, morphology and mechanics in septic patients correlate with disease severity [9]. We therefore hypothesized that the T-cell response to bacterial virulence factors or living bacteria is also concentration-dependent. To test this hypothesis, primary T-cells were exposed to different concentrations of culture supernatant derived from *S. aureus* strain 6850 (Figure 4A,C) or infected with different amounts of living bacteria (Figure 4B). As expected, the changes in T-cell morphology strongly depended on the concentration under both conditions. However, this effect was more pronounced when the cells were treated with sterile culture supernatant compared to living bacteria. This might result from our experimental procedure. In order to prevent bacterial overgrowth and interference with the DHM measurement, we removed the bacteria four hours after infection of the cells. The cellular response under these conditions might therefore be weaker compared to the supernatant treatment.

Taken together, we were able to show that the T-cell response upon bacterial challenge depends on the causative pathogen and the concentration of the bacterial determinant. Alterations of morphological parameters such as single-cell area, cell shape or in quantitative phase contrast can be captured using DHM. However, our study has some limitations. Despite the use of primary T-cells, our cell culture approach can only partly reflect the in vivo situation. The immune response to infection and development of sepsis is based on a complex interaction of different cell types, including direct cell contacts and production of various cytokines [65]. However, T-cells themselves also express pattern recognition receptors, e.g., toll-like receptors, and can be directly activated by virulence factors [66,67]. We therefore assume that our results relate to the in vivo setting to a certain degree, and T-cells from sepsis patients are morphologically different compared to healthy controls. Previous studies using hematology analyzers described significant alterations of monocyte and neutrophil cell sizes under septic conditions compared to healthy controls [7,18,68]. In our future studies, we aim to discriminate not only between infection and sepsis but also between different causative agents. Thereby, we want to investigate the potential of DHM in the context of sepsis diagnosis and identification of the underlying bacteria. A correct and rapid diagnosis along with early administration of pathogen-specific antibiotics is crucial to improve patient outcome [69,70]. For conventional blood culture procedures, it is necessary to grow the bacteria, which is time-consuming and can take up to 72 h. Moreover, blood cultures are often negative, impeding a targeted treatment of the patient. If we are able to transfer our findings into the clinical setting, application of QPI by DHM could rapidly provide useful information about the infecting agent and guide the antibiotic treatment. DHMs are commercially available and could be easily implemented into the clinical workflow. No staining or labeling procedures are applied, making it is a fast and easy-to-use method that does not require extensive training. Moreover, QPI provides quantitative information, allowing for the integration of machine-learning models to improve and facilitate data interpretation. DHM could rather complement established diagnostic tools than replace them. Evaluation of immune cell morphology cannot reach the same accuracy such as bacteriology techniques or provide information about antibiotic resistances. Additional tests, such as antibiotic susceptibility testing, are still needed for a precise diagnosis. However, QPI could provide first information to support an early treatment initiation that is followed by the more time-consuming methods for further specification. Our study indicates that analysis of T-cell morphology with DHM might be a promising approach to support such an early identification of sepsis patients.

## Figures and Tables

**Figure 1 cells-12-00762-f001:**
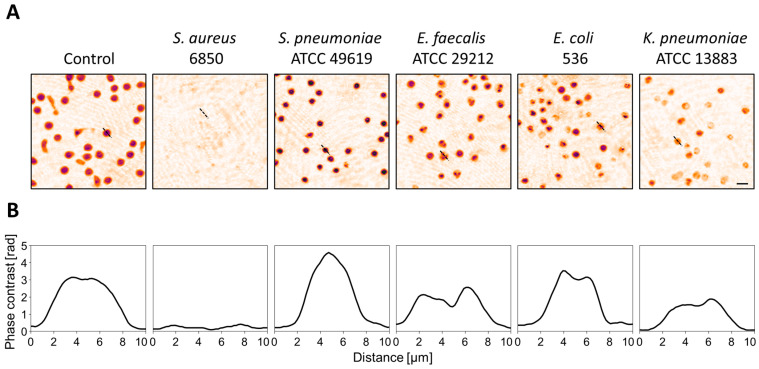
Distinct cell morphologies after exposure to bacterial culture supernatant derived from different species. Human primary T-cells were treated with sterile bacterial culture supernatants derived from various species or left untreated as control. DHM was applied to visualize morphological changes. Representative images illustrating the cell morphology 10 h after the addition of supernatants are shown. (**A**) Pseudo-colored phase contrast images. The scale corresponds to 10 µm. (**B**) Distribution of phase contrast of cross-sections through single cells (black dashed line in A).

**Figure 2 cells-12-00762-f002:**
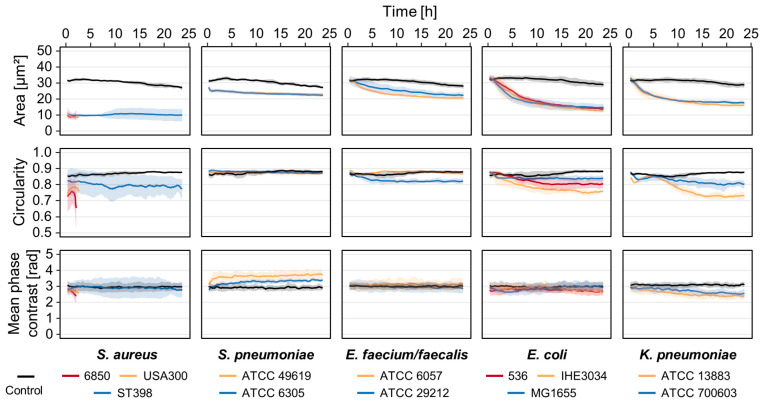
T-cell response to bacterial culture supernatant depends on both the species and strain. Human primary T-cells were exposed to sterile bacterial culture supernatants from various species and strains (colored lines) or left untreated as control (black lines). Timelapse DHM was applied to monitor the cellular response over time, and the resulting quantitative phase contrast images were analyzed for average single cell area, average circularity and mean phase contrast per cell. Results are presented as mean (solid lines) ± SD (shading) of at least three independent experiments. The curves were smoothed using moving averages with a window size of 15 measuring points.

**Figure 3 cells-12-00762-f003:**
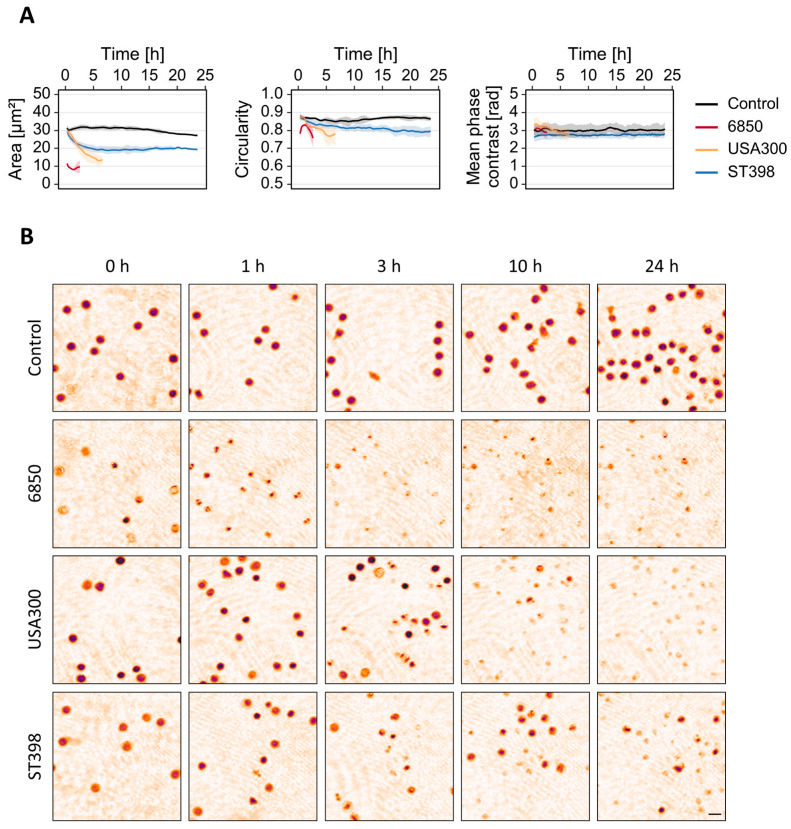
*S. aureus* MVs induce rapid changes in T-cell morphology that vary between different strains. Human primary T-cells were treated with MVs (2 × 10^9^ particle/mL) derived from different *S. aureus* strains or left untreated as control. The phenotypical cell response was recorded over 24 h by timelapse DHM. (**A**) Average cell area, average circularity and mean phase contrast per cell were analyzed using quantitative phase contrast images. Results represent mean (solid lines) ± SD (shading) of at least three independent biological replicates. The curves were smoothed using moving averages with a window size of 15 measuring points. (**B**) Representative pseudo-colored phase contrast images at indicated measurement time points. The scale bar corresponds to 10 µm.

**Figure 4 cells-12-00762-f004:**
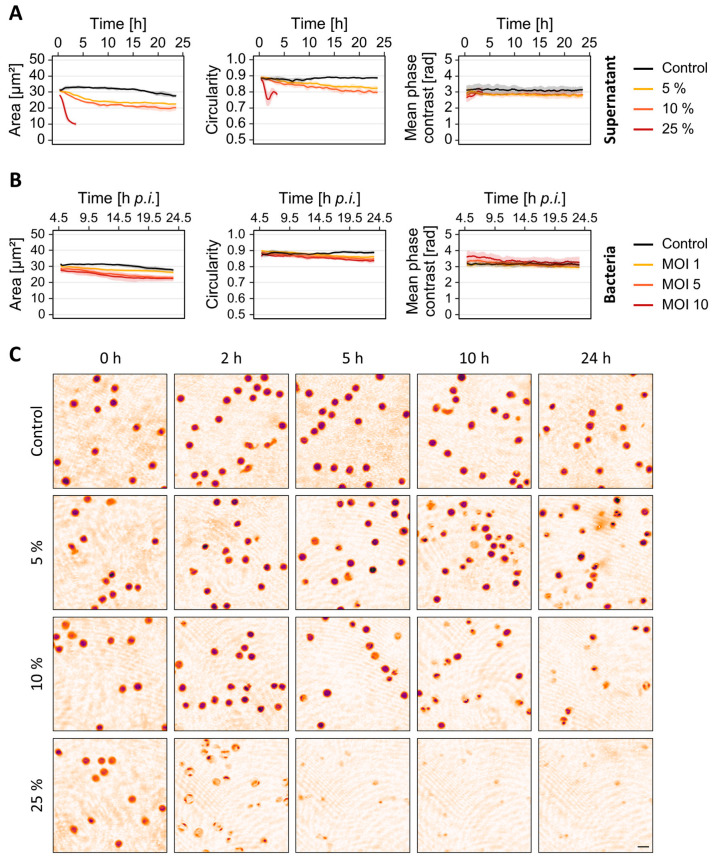
Concentration-dependent effect of bacterial culture supernatant and living bacteria on T-cell morphology. (**A**) Human primary T-cells were exposed to different concentrations of *S. aureus* 6850 culture supernatant. (**B**) T-cells were infected with different bacterial loads of *S. aureus* 6850. Timelapse DHM was started 5 h after the addition of bacteria. (**A**,**B**) Quantitative phase contrast images were analyzed for average single cell area, average circularity and mean phase contrast per cell. Results are presented as mean (solid lines) ± SD (shading) of at least three independent measurements. The graphs were smoothed using moving averages with a window size of 15 measuring points. (**C**) Representative pseudo-colored phase contrast images of T-cells treated with *S. aureus* 6850 supernatant at indicated measurement time points. The scale bar corresponds to 10 µm.

**Table 1 cells-12-00762-t001:** Bacterial strains used in this study.

Species	Strain	Reference
*S. aureus*	6850	[33]
	USA300	[34,35]
	ST398	[36]
*S. pneumoniae*	ATCC 49619	ATCC 49619
	ATCC 6305	ATCC 6305
*E. faecium*	ATCC 6057	ATCC 6057
*E. faecalis*	ATCC 29212	ATCC 29212
*E. coli*	536	[37]
	IHE3034	[38]
	MG1655	[39]
*K. pneumoniae*	ATCC 13883	ATCC 13883
	ATCC 700603	ATCC 700603

## Data Availability

Not applicable.

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
