# Peer review of "Application of Digital Holographic Microscopy to Analyze Changes in T-Cell Morphology in Response to Bacterial Challenge"

_cells, 2023, doi:10.3390/cells12050762_

Round 1

Reviewer 1 Report

This manuscript capitalizes on the expertise and experience of the investigators with the application of digital holographic microscopy to the study of lymphocytes and builds on a previous publication by the same group of investigators (Microorganisms 2022, 10, 391. Investigating Morphological Changes of T-lymphocytes after Exposure with Bacterial Determinants for Early Detection of Septic Conditions.)

The major difference between the present study and the earlier one is the use of human T-lymphocytes isolated from whole blood samples of normal volunteers.

The idea and intent of exploring applying this methodology to the study of human lymphocytes in the study of diseases is well received and exciting. I commend the objectivity of the authors own assessment of the results in the broader context of the complexity of cell-interactions that underlie cellular immune responses. They openly and explicitly acknowledge the limitations of their study design in lines 333-338 of the discussion section of their manuscript. Given such limitation they objectively limit their conclusions on their results and purposely constrain them to the obvious expectation of alterations in cell size, mean cell contrast (multiple factors account this effect) and cellular integrity (apoptosis). The experiments to examine cellular responses to bacterial challenges are well conceived, described, presented and discussed in the manuscript.

I read this manuscript with great interest because of the study of living cells expecting to produce inferences about cell morphology from physical variables examined across time, hoping to learn more about the dynamics of cellular immune responses. But the actual work and the results from the experimental design does not allow it. Nevertheless, the work is well executed and presented, as it was in their previous published work using Jurka cells (TIB-152; ATCC, Manassas, USA).

Final comments:

The exposure of T-lymphocytes to supernatants of Monocytes exposed to the same bacterial lines and determinants would have provided results more closely related to the current understanding of the cellular immune responses. Particularly of T-cells whose receptors interact with the determinants pre-processed by Macrophages and presented on their surface membranes in the context of HLA complexes.

Do membrane vesicles carry these pre-processed bacterial determinants? I would expect them to do so and it would have been an exciting observation on cellular immune responses to bacterial antigens providing a dynamic and innovative research perspective with potential for clinical exploration in septicemia and understanding the evolution of the immune dysregulation leading to sepsis. This comment is well intended and meant to be constructive and stir the obvious passion of the researchers in exploring this methodological approach to the study alterations of cell morphology in disease states. This should also broaden the application of this interesting and exciting approach to produce dynamic measures of changes in cellular morphology in living cells that could be extrapolated to improvements in instrumentation in clinical hematology laboratories.

Overall the work is well executed, the methods are strong and well described as are the results on graphs and figures. I missed at least one figure of cell morphological changes using conventional hematological staining of cell preparations  showing the time-lapse alterations of cell morphology.

These may have allowed a classical assessment of altered cell morphology and perhaps recognized evolving stages of cell death by apoptosis.

Reviewer 2 Report

This manuscript applies QPI to measure morphological difference in primary T cells during exposure to bacteria. It presents intriguing results showing differential responses of T cells to supernatant from bacterial cultures. This manuscript is generally well reported, however, a number of methods need to be clarified and expanded upon prior to publication.

1. The T cells used here were isolated from healthy human volunteers. How was potential variability from patient to patient accounted for? E.g. if the strain-dependent results (Fig. 3) were performed on T cells from different patients, then the observed varaibility in responses may be due to variation in patient-specific susceptibility to bacterials MVs.

2. Related to my first question: how many patient samples were used? How many cells were imaged per trial. E.g. figure captions do not state number of realizations. Also, in the context of this study, what is meant as a 'biological replicate'?

3. How do the results of this study compare to white blood cell changes observed in other studies, e.g. ref 9, 7, 18, 63? Even though this study used QPI, are these results generally consistent with this previous work?

4. Is there a dose dependent response to MVs? how was 2e9 particles/mL chosen as the appropriate dose? Is this a relevant physiological value during sepsis?

5. In discussion: it would be nice to include how does the QPI approach potentially compares (timing, accuracy, applicabilit to unknown samples, etc.) to standard methods for diagnosis (e.g. culturing based methods). Where can QPI do better? Where would it be worse? The last paragraph starts to mention this, but does not really outline how this approach could be used clinically.

6. Opinion: Figure 1C does not add anything to the presentation of results and should be omitted. The pseudo 3D view shows no more (and potentially less!) information than the pseudo-colored images in panel A.

Round 2

Reviewer 2 Report

The authors have responded sufficiently to my concerns. I have no additional concerns.